# Advanced Modeling and Signal Processing Methods in Brain–Computer Interfaces Based on a Vector of Cyclic Rhythmically Connected Random Processes

**DOI:** 10.3390/s23020760

**Published:** 2023-01-09

**Authors:** Serhii Lupenko, Roman Butsiy, Nataliya Shakhovska

**Affiliations:** 1Faculty of Electrical Engineering, Automatic Control and Informatics, Opole University of Technology, 45-758 Opole, Poland; 2Institute of Telecommunications and Global Information Space, National Academy of Sciences of Ukraine, 02000 Kyiv, Ukraine; 3Institute of Computer Sciences and Information Technologies, Lviv Polytechnic National University, 79000 Lviv, Ukraine

**Keywords:** brain–computer interface systems, electroencephalographic signals, vector of cyclic rhythmically connected random processes, compatible statistical analysis

## Abstract

In this study is substantiated the new mathematical model of vector of electroencephalographic signals, registered under the conditions of multiple repetitions of the mental control influences of brain–computer interface operator, in the form of a vector of cyclic rhythmically connected random processes, which, due to taking into account the stochasticity and cyclicity, the variability and commonality of the rhythm of the investigated signals have a number of advantages over the known models. This new model opens the way for the study of multidimensional distribution functions; initial, central, and mixed moment functions of higher order such as for each electroencephalographic signal separately; as well as for their respective compatible probabilistic characteristics, among which the most informative characteristics can be selected. This provides an increase in accuracy in the detection (classification) of mental control influences of the brain–computer interface operators. Based on the developed mathematical model, the statistical processing methods of vector of electroencephalographic signals are substantiated, which consist of statistical evaluation of its probabilistic characteristics and make it possible to conduct an effective joint statistical estimation of the probability characteristics of electroencephalographic signals. This provides the basis for coordinated integration of information from different sensors. The use of moment functions of higher order and their spectral images in the frequency domain, as informative characteristics in brain–computer interface systems, are substantiated. Their significant sensitivity to the mental controlling influence of the brain–computer interface operator is experimentally established. The application of Bessel’s inequality to the problems of reducing the dimensions (from 500 to 20 numbers) of the vectors of informative features makes it possible to significantly reduce the computational complexity of the algorithms for the functioning of brain–computer interface systems. Namely, we experimentally established that only the first 20 values of the Fourier transform of the estimation of moment functions of higher-order electroencephalographic signals are sufficient to form the vector of informative features in brain–computer interface systems, because these spectral components make up at least 95% of the total energy of the corresponding statistical estimate of the moment functions of higher-order electroencephalographic signals.

## 1. Introduction

For more than half a century, research has been conducted in the direction of building effective, noninvasive brain–computer interfaces, which are based on the registration and processing of electroencephalographic (EEG) signals, from the surface of the scalp of the brain–computer interface (BCI) operator. BCI technologies have been used to solve numerous applied problems of controlling external objects in various spheres of human activity. In particular, these technologies have been effectively applied for controlling a mobile robot [1], automatic robot arm [2], artificial knee prosthesis [3], artificial hand prosthesis [4], a Mental Pool Game powered by Unity3D [5], many other video games [6], controlling a wheelchair [7], game design [8], modeling of 3D objects [9], detecting facial expressions [10], motor imagery classification [11], and imagined vowels classification [12].

The effectiveness of the functioning of noninvasive BCI-systems can be assessed by the accuracy (reliability) and computational complexity of the corresponding algorithms for processing EEG signals. The higher the accuracy and the lower the computational complexity of these algorithms, the more effective they are. Despite the existence of many approaches, software and hardware tools, mathematical models, and methods for processing EEG signals in noninvasive BCI-systems, the accuracy of the detection and recognition of the class of mental control signals of the BCI operator remains low. Additionally, the computational complexity of the BCI system algorithms remains quite high, which is due to the high computational complexity of the algorithms for training the classifiers of these systems. Actually, this state of affairs is the main motive for conducting active scientific research and development of the latest BCI technologies, which is currently observed in many scientific publications on this topic.

It is known that the procedure for processing EEG signals in BCI systems can be divided into five main stages (see Figure 1): signal registration, preprocessing of signals, evaluation of signal characteristics, classification (recognition) of signals, and computer interaction.

Each of these main stages of the EEG signal processing procedure has its own contribution to both the accuracy and computational complexity characteristics of the functioning of BCI systems. However, the accuracy and computational complexity of the procedure for EEG signal processing in BCI systems is mainly determined by the accuracy and computational complexity of the procedures for evaluating characteristics and detecting signals (classification). In turn, the procedure for evaluating the characteristics of EEG signals, the purpose of which is to obtain the informative characteristics of the EEG signals, directly affects the accuracy and computational complexity of the functioning of the detector (classifier) of the BCI system. Overall, the more sensitive the informative characteristics of the EEG to the mental controlling influences of the BCI operator, the higher the accuracy of detection and recognition of the type of control signals in BCI systems. Conversely, the smaller the dimension of the vector of informative features, which are evaluated according to the registered EEG signals, the smaller the dimension of the optimization problems, which are the most computationally complex in the classifiers used in BCI systems.

In this context, one of the possible ways to increase the accuracy and reduce the computational complexity of the algorithms in the functioning of BCI systems is the search for (selecting) the most informative (sensitive) characteristics of the EEG signals and substantiation of the minimum dimension of the vector of informative features in BCI systems. This affects the improvement in the accuracy of the functioning of neurointerface systems; additionally, due to the rejection of noninformative characteristics and features, it reduces the level of computational complexity of the algorithms used for the functioning of such systems. This problem can be solved through comprehensive research of the various characteristics (from a predetermined set) of EEG signals from the point of view of their informativeness and minimization of the dimension of the vectors of informative features determined on their basis.

The studied characteristics of EEG signals and the methods of their evaluation are explicitly or implicitly based on the corresponding mathematical model of EEG signals. Namely, the effectiveness (accuracy and computational complexity) of EEG signal processing and computer simulation of signal transformations in BCI systems depends on the quality (adequacy and constructiveness) of the mathematical model of EEG signals. One of the most important properties of a mathematical model of EEG signals is its adequacy for the temporal structure of the investigated signals and for the tasks of signals detecting (classification) in BCI systems. Another important property of the mathematical models of EEG signals is their ability to provide the researcher with the widest possible set of potential characteristics of EEG signals, among which the most informative characteristics can be selected. Based on the selected informative characteristics, a vector of the informative features with the smallest dimensions is formed, which is then fed to the input of the detector (classifier) of the BCI system (see Figure 2).

The main goal in this study was the justification and development of a new mathematical model and methods of statistically evaluating the characteristics of EEG signals, which would enable an increase in accuracy in the detection (classification) of the mental control influences of BCI operators, reducing the computational complexity of the algorithms of BCI systems.

Taking into account the essential role of the mathematical model of EEG signals as the basis for methods of their processing in ensuring high accuracy and low computational complexity of BCI system algorithms, in the next section, we provide a brief review and analysis of the literature related to the mathematical models of EEG signals and their characteristics as informative features in BCI systems.

## 2. Related Work

There are many mathematical models of EEG signals. Some mathematical models of EEG signals were developed not for the needs of detecting (classifying) the mental control influence of a BCI operator, but for other tasks such as medical diagnosis and research of physiological characteristics of human brain activity. Among the mathematical models are those describing the neurophysiological mechanisms of EEG signal generation, as well as those that describe the regularities of the temporal structure of EEG signals. Models that describe the temporal structure of EEG signals are often used both for medical diagnostic tasks and neurophysiological studies for EEG signals, and for describing EEG signals for the needs of developing BCI systems. From the point of view of the orientation of modeling procedures for different types of processing of EEG signals, two classes of mathematical models of EEG signals can be distinguished. The first class includes the mathematical models of EEG signals, because they are mainly used only for solving problems of preprocessing (mainly filtering) of EEG signals, and not for evaluating the characteristics of EEG signals. The second class of mathematical models includes models directly related to the tasks of evaluating the information in EEG signals.

For example, the models and methods were developed for the removal of electrooculography and electromyography artifacts [13] and for eye movement artifacts [14], and a review of various other methods for the removal of artifacts from EEG signals [15] was carried out. For problems of filtering EEG signals from noise [16] and independent component analysis [17], the least-squares method were used. These methods are based on an additive mathematical model of EEG signals, which includes two components, namely, a deterministic function (for example, a quasiharmonic function that corresponds to the result of amplitude and angular modulation of a harmonic carrier) and a stationary random process (often stationary white noise is used). Similar additive models were used [18] for smoothing filtering of EEG signals and for the development and application of smoothing filter based upon the Savitzky–Golay filter for preprocessing of EEG signals [19] Researchers [20] presented the results of the automated detection of small no-brain artifacts by using higher-order statistics and independent component analysis of EEG signals. Others [21] reported the results of automated detection of artifacts of different origins by using independent component analysis of EEG signals, where their mathematical model was an additive model containing noise with either sub- or super-Gaussian distributions.

A study [22] was dedicated to mathematical deterministic models of EEG signals, researching models of the neurophysiological generation of EEG signals based on Maxwell equations. Similar approaches to the modeling of neurophysiological mechanisms of formation of EEG signals were considered [23], which was dedicated to solving the inverse problem in EEG source analysis. Additionally, authors [24] presented the deterministic mathematical models describing the neurophysiological mechanisms of EEG signal generation at the micro, meso, and macro levels of human brain organization. Several deterministic modeling approaches, which are the basis of the various EEG decoding algorithms and classification methods, were considered in a review [25].

Within the framework of the deterministic approach to modeling EEG signals (EEG signals are interpreted as deterministic functions), methods of their harmonic analysis, as well as methods of spectral analysis in nonharmonic bases, are used. For example, a new method for the spatial harmonic analysis of EEG data using the Laplacian eigenspace of the meshed surface of electrode positions was developed [26]. In a study [27], EEG signals were explored using wavelets analysis for the purpose of feature extraction and classification by the application of an artificial neural network and support vector machine. In another study [28], for detection time-localized events in the structure of EEG signals, wavelet analysis was used. In a study [29], sub-band wavelet entropy and its time difference were proposed as two quantitative measures for analyzing and segmenting EEG signals.

In addition to deterministic mathematical models of EEG signals, stochastic mathematical models are actively used in the form of random processes and vectors of random processes. Within the framework of the stochastic approach, fractal EEG models are used. For example, authors [30] used fractional Gaussian noise and fractional Brownian motion for the decomposition of EEG signals into brain rhythms: delta, theta, alpha, beta, and gamma rhythms. In a study [31], multifractal EEG analysis was used for the task of emotion recognition.

Among the stochastic mathematical models of EEG signals, a stationary random process is often used in the general sense (informative characteristics, autocorrelation function, and spectral function of the power density (power spectrum) of the EEG signal). For example, as an informative characteristic of EEG signals, researchers [32] used the one-dimensional power spectrum, which corresponds to the model of EEG signals in the form of a stationary random process.

The Markov process was used to model and computer simulate EEG signals tp identify pathophysiological EEG changes in clinical diagnosis [33]. In a study [34], an autoregression model and the Yule–Walker algorithm were used for the analysis and forecasting of EEG signals (informative characteristics and coefficients of the of autoregression models). A similar approach was considered [35], where stochastic differential equations were used to model EEG signals. In a study [36], both linear and nonlinear stochastic mathematical models of EEG signals were used. A study [37] was devoted to linear models of EEG signals for linear discriminant, principal component, and independent component analyses. Stationary linear random processes (informative characteristics and parameters of the kernel of the linear process and probabilistic characteristics of the generating process with independent increases) [38] and stationary conditional linear random processes [39] were used for the analysis of EEG signals, which made it possible to take into account the neurophysiological processes of generating EEG signals. Others [40] used stochastic partial differential equations for modeling mesoscopic electrical activity of the human cortex. A similar approach to modeling was used in another study [41].

To study the correlations between the EEG signals obtained from different electrodes in the context of the news vendor problem, a model of EEG signals was used in the form of a vector of stationary and stationary-connected random processes [42]. Within this model, autocorrelation, cross-correlation, and power density spectral functions were used as informative characteristics of the vector EEG components. To take into account the nonstationary (namely, the piecewise stationary) nature of the EEG time structure, the piecewise stationary random process (informative characteristics and autocorrelation functions and spectral functions of the power density of the EEG signal at different time intervals of registration) was used as an EEG signal model. In particular, in [43], a piecewise stationary random process was used as a model of EEG signal for its adaptive segmentation. Additionally, in [44], this mathematical model was used to solve the problem of spectral analysis of EEG signals in short time intervals.

In a study [45], the cyclostationary correlated (periodically correlated) random process, as a mathematical model of EEG signals, for use in brain–computer interface design was justified. Within this mathematical model, moment functions of the first and second orders and the bispectral power density function of the EEG signals were used as informative characteristics. Often, for estimating the probabilistic characteristics of EEG signals, methods of averaging by an ensemble of realizations are used for nonstationary random processes with an undefined probabilistic structure. For example, the EEG signal was presented through an ensemble of its implementations [1], where the authors proposed using the results of the integration of EEG signals during each second as informative characteristics.

Despite the significant number of mathematical models of EEG signals, these mathematical models, with the exception of the cyclostationary correlated random process, do not take into account the cyclic structure of EEG signals registered under the conditions of multiple repetitions of the mental control influences of the BCI operator during the stage of classifier training. Additionally, the cyclostationary correlated random process cannot take into account moment functions of a higher order or the variability in the rhythm of the characteristics of investigated signals; it does not have formal means of displaying the relationships between the components of the vector EEG signal. The well-known methods of evaluating an ensemble of recorded EEG signal cycles do not take into account the relationships between the characteristics of different EEG signal cycles in their multicycle realization. Notably, most of the methods for evaluating the informative characteristics of EEG signals are methods for evaluating the moment functions of EEG signals within the framework of the spectral-correlation theory of random processes only, and not the theory of random processes within the framework of multidimensional distribution functions and higher-order moment functions.

The disadvantages described above are possible reasons for the low accuracy of detection and classification of the controlling mental instructions of the BCI operator, which indicates the need to improve the mathematical models and methods for evaluating the informative characteristics of EEG signals in BCI systems.

The objective of this study was the development of new mathematical model, methods, and software tools for statistical processing of EEG signals in BCI systems that enable comprehensive research on EEG signals for the purpose of searching for and selecting the most informative (sensitive) characteristics of the EEG signals and determining the minimum vector dimensions of the informative features in BCI systems. The results of a comparative analysis of known mathematical models and the mathematical models of EEG signals we developed in the study are summarized in Table 1.

## 3. The New Mathematical Model of EEG Signals in BCI Systems in the Form of a Vector of Cyclic Rhythmically Connected Random Processes with Two Zones on Cycles

Given the goal of the study, a system of requirements was formulated for the new mathematical model of EEG signals in BCI systems we developed.

As a result of many uncontrollable natural and artificial factors, and considering the results of many scientific studies devoted to the modeling and analysis of EEG signals, it was appropriate to use the stochastic approach for building a mathematical EEG model. As EEG registration is carried out from *N*
(N≥2) different electrodes (leads) on the surface of the operator’s head, *N* synchronously registered EEG signals were obtained in the experiments (this collection of *N* EEG signals is called the vector EEG); then, it was reasonable to assume that the general mathematical model of vector EEG is an *N*-dimensional vector of random processes.

In order to train the system for detecting (classifying) the controlling mental influence of the BCI operator on the structure of the vector EEG, we conducted a series of *M* identical experiments. A series of training experiments is a series of mental acts of the BCI operator, which are manifested in changes in the temporal structure of the vector EEG and are the basis for training the software subsystem to recognize this act of mental influence of the operator on the object of manipulation, by analyzing the vector EEG. In addition, this multicycle vector EEG can be used as primary statistics to study the patterns of the training and effectiveness of BCI operators.

If, as a result of the mental action of the BCI operator, a change occurs in the temporal (or spectral) structure of the vector EEG that is significant for solving detection (classification) problems; then, as a result of conducting a series of *M* time-consecutive experiments of the same type (mental effects of the same operator under the same conditions), it is reasonable to assume a certain cyclicity and repeatability of the time structure of the vector EEG, which reflects the corresponding control information in the BCI system. In the absence of such a series of sequential mental influences of the BCI operator, the cyclic structure of the vector EEG is absent. Because the expediency of the stochastic approach to the mathematical modeling of EEG signals was formulated above, such cyclic repeatability should be present in the probabilistic structure (probabilistic characteristics) of the investigated vector EEG.

The presence of a cyclic structure in the probabilistic characteristics of the vector EEG, which reflects the electrical activity of the brain of the BCI operator in a series of consecutive identical experiments, is the basis for the development of methods and means of detecting and classifying control information in BCI systems. In view of the above, it is reasonable to demand the mathematical model of EEG signals to reflect this cyclic repeatability in probabilistic characteristics.

Additionally, as each cycle of the vector EEG can be divided into zones of activity (when the operator performs a mental control action) and passivity (when the operator’s mental control action is absent), the mathematical model must take into account this zone structure in agreement with its cyclical structure.

The presence of cyclicity in certain characteristics of the vector EEG is the basis of the possibility of solving the task of detecting the control signals of the BCI operator, because only in this case is the nonstationary structure of the EEG signals suitable for effectively distinguishing the areas that correspond to the zones of activity and passivity of the BCI operator. The cyclical nature of vector EEG characteristics is a necessary condition for successfully solving a more complex task than detecting the control signal of the BCI operator, namely, the task of recognizing the type of control signal of the BCI operator, because, before classifying the control signal, it must be detected and distinguished from the EEG stream data.

The durations of zones of activity and passivity within different cycles of vector EEG, and as a consequence the duration of its different cycles in general, are different, so the mathematical model must take into account this temporal variability. As a vector EEG is an ordered set of EEG signals from different electrodes that reflect the electrical activity of different areas of the brain of the same person (operator), and the time intervals during which the operator carries out their mental act for all components of the vector EEG are the same, then it is reasonable to postulate a rhythmic connection of all components of the vector EEG.

It is also important that the mathematical model of EEG signals can ensure the integration of statistical information from EEG signals obtained from different sensors; in particular, it allows conducting a joint statistical analysis of EEG signals from different brain regions of the BCI operator. In particular, in addition to the structural similarity of EEG signals and their mathematical model, an important property of such a model is its ability to describe the widest possible class of characteristics of the studied signals in order to identify those most sensitive to the effects of the BCI operator.

Taking into account the above requirements for the mathematical model of EEG signals, as an adequate mathematical model of EEG signals, it is appropriate to use a vector of cyclic rhythmically connected random processes. The vector of cyclic rhythmically connected random processes is defined according to the literature [46].

**Definition** **1.**
*Vector $\Theta_{N}\left(\omega ,t\right)$ of random processes $\left\{\xi_{i}\left(\omega ,t\right),i=\bar{1,N},\omega \in \Omega ,t\in R\right\}$ is called*
***the vector of cyclic rhythmically connected random processes**
(and the processes themselves are called*
*
**the cyclic rhythmically connected random processes**
*
*), if there exists such a function T(t,n),t∈R,n∈Z, which satisfies conditions (1)–(3) of the rhythm function and, for any $t_{1},\ldots ,t_{k}$ from the set of separability of vector $\Theta_{N}\left(\omega ,t\right)$, k-dimensional random vectors $\left\{\xi_{i_{1}}\left(\omega ,t_{1}\right),\xi_{i_{2}}\left(\omega ,t_{2}\right),\ldots ,\xi_{i_{k}}\left(\omega ,t_{k}\right)\right\}$ and $\left\{\xi_{i_{1}}\left(\omega ,t_{1}+T\left(t_{1},n\right)\right),\xi_{i_{2}}\left(\omega ,t_{2}+T\left(t_{2},n\right)\right),\ldots ,\right.$$\left.\xi_{i_{k}}\left(\omega ,t_{k}+T\left(t_{k},n\right)\right)\right\},n\in Z,i_{1},\ldots ,i_{k}=\bar{1,N}$ are stochastically equivalent in the wide sense for all n∈Z and for all k∈N.*


Function T(t,n),t∈R,n∈Z in [47] is called a ***rhythm function*** if it has the following properties:

(1)
(1)$$\left\{\begin{array}{c} T(t,n)>0\;(T(t,1)<\infty ),t\in R,\;if\;n>0,\\ T(t,n)=0,\;t\in R,\;if\;n=0,\\ T(t,n)<0,\;t\in R,\;if\;n<0. \end{array}\right.$$

(2) For any $t_{1}\in R$ and $t_{2}\in R$ for which $t_{1}<t_{2}$ and for function T(t,n), a strict inequality holds:(2)$$T\left(t_{1},n\right)+t_{1}<T\left(t_{2},n\right)+t_{2},\forall n\in Z;$$

(3) Function T(t,n) is the smallest in modulus $\left(\left|T(t,n)\right|\leq \left|T_{\gamma }\left(t,n\right)\right|\right)$ among all such functions $\left\{T_{\gamma }\left(t,n\right),\gamma \in N\right\}$, which satisfy (1) and (2), namely:(3)$$\left|T(t,n)\right|=min_{\gamma \in N}\left\{\left|T_{\gamma }\left(t,n\right)\right|,\gamma \in N\right\},t\in R,n\in Z.$$

Note that stochastically equivalent in the wide sense is *k*-dimensional random vectors, if they are the same *k*-dimensional distribution functions.

In the case where the vector $\Theta_{N}\left(\omega ,t\right)$ contains only one component (N=1), then such a component is a cyclic random process ξ(ω,t),ω∈Ω,t∈R [48]. In the partial case when T(t,n)=n·T,T=const, T>0, the vector of cyclic rhythmically connected random processes is the vector of T-periodic and *T*-periodically connected random processes (*T*-periodic random vector). The rhythm function T(t,n),t∈R,n∈Z determines the law of changing the time intervals between the single-phase values of the vector of cyclic rhythmically connected stochastic processes. The rhythm of the cyclic signal in qualitative terms can be regular (stable and unchanging) or irregular (variable and unstable). From the point of view of the concept of rhythm function, the vector of periodic and periodically connected random processes is a the vector of cyclic rhythmically connected random processes with a regular (stable) rhythm or, rather, with a rhythm function T(t,n)=n·T,T=const>0. The vector of irregular rhythm cyclic signals (nonrhythmic cyclic signals and variable rhythm signals) are signals whose model is a vector of cyclic rhythmically connected random processes with a rhythm function T(t,n)≠n·T(T(t,1)≠const). Such a vector of cyclic rhythmically connected random processes is called a vector of cyclic rhythmically connected random processes with an irregular (variable) rhythm. The vector of cyclic rhythmically connected random processes and cyclic random process are widely used as mathematical models of the synchronously registered cardio signals that differ in physical nature [49], the processes of multiple cracking of nanocoating [50], and the processes of ordering surface structures of materials [51]. These mathematical models and methods of signal processing have been represented in the form of a taxonomic tree [52] and an ontology [53] in onto-oriented expert support systems for model decision making.

Because the duration of an operator’s performance of the corresponding task in each experiment is different, the corresponding durations of the cycles of vector of cyclic rhythmically connected random processes are also different. This indicates the correctness of the assumption that the rhythm function of the vector of cyclic rhythmically connected random processes does not satisfy the condition T(t,n)=n·T. Thus, taking into account the above considerations, it is reasonable to assume that an adequate mathematical model of a vector EEG is a vector of cyclic rhythmically connected random processes with an irregular rhythm.

Let us consider the properties of some probabilistic characteristics of vector $\Theta_{N}\left(\omega ,t\right)$ of cyclic rhythmically connected random processes. Thus, for its compatible *k*-dimensional distribution function $F_{k_{\xi_{i_{1}}\ldots \xi_{i_{k}}}}\left(x_{1},\ldots ,x_{k};t_{1},\ldots ,t_{k}\right)$, there is the equality:(4)$$\begin{array}{c} F_{k_{\xi_{i_{1}}\ldots \xi_{i_{k}}}}\left(x_{1},\ldots ,x_{k};t_{1},\ldots ,t_{k}\right)=F_{k_{\xi_{i_{1}}\ldots \xi_{i_{k}}}}\left(x_{1},\ldots ,x_{k};t_{1}+T\left(t_{1},n\right),\ldots ,t_{k}+T\left(t_{k},n\right)\right),\\ i_{1},\ldots ,i_{k}=\bar{1,N},t_{1},\ldots ,t_{k}\in R,n\in Z,k\in N. \end{array}$$

If there is a compatible density distribution function of $\Theta_{N}\left(\omega ,t\right)$, then it has an equality that is similar to equality (4).

Compatible *k*-dimensional characteristic function $f_{k_{\xi_{i_{1}}\ldots \xi_{i_{k}}}}\left(u_{1},\ldots ,u_{k};t_{1},\ldots ,t_{k}\right)$ of vector $\Theta_{N}\left(\omega ,t\right)$ satisfies the equality:(5)$$\begin{array}{c} f_{k_{\xi_{i_{1}}\ldots \xi_{i_{k}}}}\left(u_{1},\ldots ,u_{k};t_{1},\ldots ,t_{k}\right)=f_{k_{\xi_{i_{1}}\ldots \xi_{i_{k}}}}\left(u_{1},\ldots ,u_{k};y\left(t_{1},n\right),\ldots ,y\left(t_{k},n\right)\right)=\\ =f_{k_{\xi_{i_{1}}\ldots \xi_{i_{k}}}}\left(u_{1},\ldots ,u_{k};t_{1}+T\left(t_{1},n\right),\ldots ,t_{k}+T\left(t_{k},n\right)\right),\\ i_{1},\ldots ,i_{k}=\bar{1,N},t_{1},\ldots ,t_{k}\in R,n\in Z,k\in N. \end{array}$$

The mixed initial moment function of order $p=\sum_{j=1}^{k}r_{j}$ of vector $\Theta_{N}\left(\omega ,t\right)$ satisfies the equality:(6)$$\begin{array}{c} C_{p_{\xi_{i_{1}}\ldots \xi_{i_{k}}}}\left(t_{1},\ldots ,t_{k}\right)=E\left\{\xi_{i_{1}}^{r_{1}}\left(\omega ,t_{1}\right)\cdot \ldots \cdot \xi_{i_{k}}^{r_{k}}\left(\omega ,t_{k}\right)\right\}=\\ =C_{p_{\xi_{i_{1}}\ldots \xi_{i_{k}}}}\left(t_{1}+T\left(t_{1},n\right),\ldots ,t_{k}+T\left(t_{k},n\right)\right),\\ t_{1},\ldots ,t_{k}\in R,i_{1},\ldots ,i_{k}=\bar{1,N},n\in Z,k\in N. \end{array}$$

The mixed central moment function of order $p=\sum_{j=1}^{k}r_{j}$ of vector $\Theta_{N}\left(\omega ,t\right)$ satisfies the equality:(7)$$\begin{array}{c} R_{p_{\xi_{i_{1}}\ldots \xi_{i_{k}}}}\left(t_{1},\ldots ,t_{k}\right)=E\left\{{\left(\xi_{i_{1}}\left(\omega ,t_{1}\right)-m_{\xi_{i_{1}}}\left(t_{1}\right)\right)}^{r_{1}}\cdot \ldots \cdot {\left(\xi_{i_{k}}\left(\omega ,t_{k}\right)-m_{\xi_{i_{k}}}\left(t_{k}\right)\right)}^{r_{k}}\right\}=\\ =R_{p_{\xi_{i_{1}}\ldots \xi_{i_{k}}}}\left(t_{1}+T\left(t_{1},n\right),\ldots ,t_{k}+T\left(t_{k},n\right)\right),\\ t_{1},\ldots ,t_{k}\in R,i_{1},\ldots ,i_{k}=\bar{1,N},n\in Z,k\in N. \end{array}$$

The above probabilistic characteristics of a vector of cyclic rhythmically connected random processes are informative; however, they are too computationally cumbersome. Therefore, to statistically process a vector EEG, it is appropriate to use less informative, but much more computationally simple, probabilistic characteristics, such as the vector of mathematical expectations and the matrix of correlation functions of vector $\Theta_{N}\left(\omega ,t\right)$.

As each cycle of the vector EEG can be divided into a zone of activity (when the operator performs a mental control action) and a zone of passivity (when the operator’s mental control action is absent), each component $\xi_{i}\left(\omega ,t\right)$ of vector $\Theta_{N}\left(\omega ,t\right)$ can be presented as follows:(8)$$\begin{array}{c} \xi_{i}\left(\omega ,t\right)=\sum_{m\in Z}\sum_{k=1}^{2}{\tilde{\xi }}_{i,m,k}\left(\omega ,t\right)=\sum_{m\in Z}\left({\tilde{\xi }}_{i,m,1}\left(\omega ,t\right)+{\tilde{\xi }}_{i,m,2}\left(\omega ,t\right)\right),\\ \omega \in \Omega ,t\in R,i=\bar{1,N}, \end{array}$$
where random process ${\tilde{\xi }}_{i,m,1}\left(\omega ,t\right)$ coincides (is identical) with random process $\xi_{i}\left(\omega ,t\right)$ in the area $W_{m,1}$ that corresponds to the state of the BCI operator’s passivity in the *m*th cycle of the cyclic random process $\xi_{i}\left(\omega ,t\right)$ and that is equal to:(9)$${\tilde{\xi }}_{i,m,1}\left(\omega ,t\right)=\xi_{i}\left(\omega ,t\right)\cdot I_{W_{m,1}}\left(t\right),\omega \in \Omega ,t\in R,i=\bar{1,N},m\in Z.$$

The random process ${\tilde{\xi }}_{i,m,2}\left(\omega ,t\right)$ coincides (is identical) with a random process $\xi_{i}\left(\omega ,t\right)$ in the area $W_{m,2}$ that corresponds to the state of the BCI operator’s activity in the *m*th cycle of the cyclic random process $\xi_{i}\left(\omega ,t\right)$ and that is equal to:(10)$${\tilde{\xi }}_{i,m,2}\left(\omega ,t\right)=\xi_{i}\left(\omega ,t\right)\cdot I_{W_{m,2}}\left(t\right),\omega \in \Omega ,t\in R,i=\bar{1,N},m\in Z.$$

Functions $I_{W_{m,1}}\left(t\right)$ and $I_{W_{m,2}}\left(t\right)$ are indicator functions:(11)$$I_{W_{m,1}}\left(t\right)=\left\{\begin{array}{c} 1,\;t\in W_{m,1},\\ 0,\;t\notin W_{m,1}, \end{array}\right.$$
(12)$$I_{W_{m,2}}\left(t\right)=\left\{\begin{array}{c} 1,\;t\in W_{m,2},\\ 0,\;t\notin W_{m,2}. \end{array}\right.$$

The union of sets $W_{m,1}$ and $W_{m,2}$ is the area of definition of the *m*th cycle of the cyclic random process $\xi_{i}\left(\omega ,t\right)$, namely:(13)$$W_{c_{m}}=W_{m,1}\cup W_{m,2},m\in Z.$$

Each *m*th cycle in the time range $W_{c_{m}}$ corresponds to the *m*th training experiment $\left(m=\bar{1,M}\right)$.

The application of a vector of cyclic rhythmically connected random processes as a model of interconnected EEG signals with the same rhythmic structure provides a number of important advantages; namely, the model takes into account:(1)The stochasticity of the vector EEG;(2)The cyclical structure of the probabilistic characteristics of the vector EEG;(3)The compatible probabilistic characteristics of various components in the vector EEG;(4)The irregularity (variability) in the rhythm of the oscillatory process;(5)The commonality of the rhythmic structure for all components of the vector EEG;(6)The stochastic dependence between different cycles of vector EEG that occur in different operator training experiments;(7)The presence of two zones (segments) in the time structure of each EEG signal, which correspond to the activity and passivity states of the BCI operator.

## 4. Methods of EEG Signal Processing in BCI Systems

The actual development of a mathematical model of a vector EEG in the form of a vector of cyclic rhythmically connected random processes with two zones in the cycles enables the development of the necessary mathematical apparatus for processing vector EEGs. Namely, the methods of estimating the characteristics of vector EEGs consist of the application of methods of statistically estimating the probabilistic characteristics of the vector of cyclic rhythmically connected random processes, and the application of methods of harmonically analyzing the obtained statistical estimates.

According to a study [47], to apply statistical methods for the estimation of probabilistic characteristics of a vector of cyclic rhythmically connected random processes, a preliminary assessment of its rhythm function is necessary. Let us consider the simplest method of estimating the rhythm function of vector EEGs, namely, a method of piecewise linear interpolation, which was previously developed [54].

Consider the simplest type of interpolation: piecewise linear interpolation. In this case, the interpolation function $\hat{T}\left(t,1\right)$ looks like:(14)$$\hat{T}\left(t,1\right)=\sum_{m\in Z}\sum_{k=1}^{2}{\hat{T}}_{mk}\left(t\right),t\in R,$$
where $\left\{{\hat{T}}_{mk}\left(t\right)\right\}$ is a set of functions equal to:(15)$${\hat{T}}_{mk}\left(t\right)=\left\{\begin{array}{c} g_{mk}\cdot t+b_{mk},t\in W_{mk},\\ 0,t\notin W_{mk},m\in Z,k=\bar{1,2}. \end{array}\right.$$

The area $W_{mk}=\left[{\tilde{t}}_{m,k},{\tilde{t}}_{m,k+1}\right]$ corresponds to the *k*th zone in the *m*th cycle. If k=2, then ${\tilde{t}}_{m,3}={\tilde{t}}_{m+1,1}$. In practice, m takes its values from a finite subset of integers.

Therefore, when piecewise-linear interpolating the rhythm function, it is necessary to find sets of coefficients $\left\{g_{mk},m\in Z,k=\bar{1,2}\right\}$ and $\left\{b_{mk},m\in Z,k=\bar{1,2}\right\}$ that completely determine the interpolation function $\hat{T}\left(t,1\right)$. We must find the expressions for calculating the required coefficients of the interpolation function. To do this, we write the equation of the segment connecting the points with the coordinates $\left({\tilde{t}}_{m,k},T\left({\tilde{t}}_{m,k},1\right)\right)$ and $\left({\tilde{t}}_{m,k+1},T\left({\tilde{t}}_{m,k+1},1\right)\right)$, which combines the readings of a discrete rhythm function $T({\tilde{t}}_{m,k},1)$ at times ${\tilde{t}}_{m,k}$ and ${\tilde{t}}_{m,k+1}$:(16)$$\frac{{\hat{T}}_{mk}\left(t\right)-T\left({\tilde{t}}_{m,k+1},1\right)}{T\left({\tilde{t}}_{m,k+1},1\right)-T\left({\tilde{t}}_{m,k},1\right)}=\frac{t-{\tilde{t}}_{m,k+1}}{{\tilde{t}}_{m,k+1}-{\tilde{t}}_{m,k}},t\in W_{mk},m\in Z,k=\bar{1,2}.$$

These equations can be reduced to the following equations:(17)$$\begin{array}{c} {\hat{T}}_{mk}\left(t\right)=\frac{T\left({\tilde{t}}_{m,k+1},1\right)-T\left({\tilde{t}}_{m,k},1\right)}{{\tilde{t}}_{m,k+1}-{\tilde{t}}_{m,k}}\cdot t+T\left({\tilde{t}}_{m,k+1},1\right)-\\ -\frac{T\left({\tilde{t}}_{m,k+1},1\right)-T\left({\tilde{t}}_{m,k},1\right)}{{\tilde{t}}_{m,k+1}-{\tilde{t}}_{m,k}}\cdot {\tilde{t}}_{m,k+1},t\in W_{mk},m\in Z,k=\bar{1,2}, \end{array}$$
or similarly:(18)$${\hat{T}}_{mk}\left(t\right)=\frac{T\left({\tilde{t}}_{m,k+1},1\right)-T\left({\tilde{t}}_{m,k},1\right)}{{\tilde{t}}_{m,k+1}-{\tilde{t}}_{m,k}}\cdot \left(t-{\tilde{t}}_{m,k}\right)+T\left({\tilde{t}}_{m,k+1},1\right),t\in W_{mk},m\in Z,k=\bar{1,2}.$$

Therefore, according to Equation (Equation 17), the coefficients $\left\{g_{mk},m\in Z,k=\bar{1,2}\right\}$ and $\left\{b_{mk},m\in Z,k=\bar{1,2}\right\}$ are determined by the ratios:(19)$$g_{mk}=\frac{T\left({\tilde{t}}_{m,k+1},1\right)-T\left({\tilde{t}}_{m,k},1\right)}{{\tilde{t}}_{m,k+1}-{\tilde{t}}_{m,k}},m\in Z,k=\bar{1,2},$$
(20)$$b_{mk}=T\left({\tilde{t}}_{m,k+1},1\right)-\frac{T\left({\tilde{t}}_{m,k+1},1\right)-T\left({\tilde{t}}_{m,k},1\right)}{{\tilde{t}}_{m,k+1}-{\tilde{t}}_{m,k}}\cdot {\tilde{t}}_{m,k+1},m\in Z,k=\bar{1,2}.$$

According to the conditions imposed on the interpolation function, its $\hat{T}\left(t,1\right)$ derivative, which exists at all points of the set R, except for the points of the set $D_{z}=\left\{{\tilde{t}}_{m,k},m\in Z,\right.$$\left.k=\bar{1,2}\right\}$ must be greater than −1. This is only possible when derivative functions $\left\{{\hat{T}}_{mk}\left(t\right)\right\}$ are greater than −1. The derivative functions $\left\{{\hat{T}}_{mk}\left(t\right)\right\}$ are equal to the coefficients $\left\{g_{mk}\left(t\right)\right\}$, which are calculated according to Formula (19) and are always greater than −1, because $T\left({\tilde{t}}_{m,k+1},1\right)-T\left({\tilde{t}}_{m,k},1\right)>{\tilde{t}}_{m,k}-{\tilde{t}}_{m,k+1}$, which follows from the conditions of the rhythm function. This is satisfied by the discrete rhythm function, $T({\tilde{t}}_{m,k},1)$, namely:(21)$$g_{mk}=\frac{T\left({\tilde{t}}_{m,k+1},1\right)-T\left({\tilde{t}}_{m,k},1\right)}{{\tilde{t}}_{m,k+1}-{\tilde{t}}_{m,k}}>-1,m\in Z,k=\bar{1,2}.$$

Let us provide the formulas for calculating the realizations of the corresponding statistical estimations of probabilistic characteristics of vector EEGs. In case of a long realization, $\Theta_{N_{\omega }}\left(t\right)=\left\{\xi_{i_{\omega }}\left(t\right),i=\bar{1,N},t\in W\subset R\right\}$ consists of M cycles. In this case, $W={⋃}_{m=1}^{M}W_{c_{m}}$.

The realization of the statistical estimation of the mathematical expectation $m_{\xi_{i}}\left(t\right)$ of each component $\xi_{i}\left(\omega ,t\right)$ of vector $\Theta_{N}\left(\omega ,t\right)$ is:(22)$${\hat{m}}_{\xi_{i}}\left(t\right)=\frac{1}{M}\cdot \sum_{n=0}^{M-1}\xi_{i_{\omega }}\left(t+T\left(t,n\right)\right),t\in W_{c_{1}}=\left[{\tilde{t}}_{1},{\tilde{t}}_{2}\right),i=\bar{1,N}.$$

The realization of statistical estimation of the dispersion $d_{\xi_{i}}\left(t\right)$ of each component $\xi_{i}\left(\omega ,t\right)$ of vector $\Theta_{N}\left(\omega ,t\right)$ is:(23)$${\hat{d}}_{\xi_{i}}\left(t\right)=\frac{1}{M}\cdot \sum_{n=0}^{M-1}{\left[\xi_{i_{\omega }}\left(t+T\left(t,n\right)\right)-{\hat{m}}_{\xi_{i}}\left(t+T\left(t,n\right)\right)\right]}^{2},t\in W_{c_{1}}=\left[{\tilde{t}}_{1},{\tilde{t}}_{2}\right),i=\bar{1,N}.$$

The realization of the statistical estimation of the initial moment function $m_{\xi_{i}}^{k}\left(t\right)$ of the *k*th order of each component $\xi_{i}\left(\omega ,t\right)$ of vector $\Theta_{N}\left(\omega ,t\right)$ is:(24)$${\hat{m}}_{\xi_{i}}^{k}\left(t\right)=\frac{1}{M}\cdot \sum_{n=0}^{M-1}\xi_{i_{\omega }}^{k}\left(t+T\left(t,n\right)\right),t\in W_{c_{1}}=\left[{\tilde{t}}_{1},{\tilde{t}}_{2}\right),i=\bar{1,N}.$$

The realization of the statistical estimation of the central moment function $d_{\xi_{i}}^{k}\left(t\right)$ of the *k*th order of each component $\xi_{i}\left(\omega ,t\right)$ of vector $\Theta_{N}\left(\omega ,t\right)$ is:(25)$$\begin{array}{c} {\hat{d}}_{\xi_{i}}^{k}\left(t\right)=\frac{1}{M-1}\cdot \sum_{n=0}^{M-1}{\left[\xi_{i_{\omega }}\left(t+T\left(t,n\right)\right)-{\hat{m}}_{\xi_{i}}\left(t+T\left(t,n\right)\right)\right]}^{k},\\ t\in W_{c_{1}}=\left[{\tilde{t}}_{1},{\tilde{t}}_{2}\right),i=\bar{1,N}. \end{array}$$

The realization of the statistical estimation of the mixed initial moment function $C_{p_{\xi_{i}}}^{k}\left(t_{1},\ldots ,t_{k}\right)$ of order $p=\sum_{i=1}^{k}r_{i}$ of each component $\xi_{i}\left(\omega ,t\right)$ of vector $\Theta_{N}\left(\omega ,t\right)$ is:(26)$$\begin{array}{c} {\hat{C}}_{p_{\xi_{i}}}^{k}\left(t_{1},\ldots ,t_{k}\right)=\frac{1}{M-M_{1}+1}\cdot \sum_{n=0}^{M-M_{1}}\left[\xi_{i_{\omega }}^{r_{1}}\left(t_{1}+T\left(t_{1},n\right)\right)-\xi_{i_{\omega }}^{r_{k}}\left(t_{k}+T\left(t_{k},n\right)\right)\right],\\ t_{1}\in W_{c_{1}},t_{2},\ldots ,t_{k}\in \overset{M_{1}}{\underset{m=1}{⋃}}W_{c_{m}},i=\bar{1,N}. \end{array}$$
where $M_{1}\;\left(M_{1}<<M\right)$ is the number of cycles in which arguments gain value $t_{2},\ldots ,t_{k}$.

The realization of the statistical estimation of a mixed central moment function $R_{p_{\xi_{i}}}\left(t_{1},\ldots ,t_{k}\right)$ of order $p=\sum_{i=1}^{k}r_{i}$ of each component $\xi_{i}\left(\omega ,t\right)$ of vector $\Theta_{N}\left(\omega ,t\right)$ is:(27)$$\begin{array}{c} {\hat{R}}_{p_{\xi_{i}}}\left(t_{1},\ldots ,t_{k}\right)=\frac{1}{M-M_{1}}\cdot \sum_{n=0}^{M-M_{1}}\left[{\left(\xi_{i_{\omega }}\left(t_{1}+T\left(t_{1},n\right)\right)-{\hat{m}}_{\xi_{i}}\left(t_{1}+T\left(t_{1},n\right)\right)\right)}^{r_{1}}\cdot \ldots \cdot \right.\\ \left.\cdot \ldots \cdot {\left(\xi_{i_{\omega }}\left(t_{k}+T\left(t_{k},n\right)\right)-{\hat{m}}_{\xi_{i}}\left(t_{k}+T\left(t_{k},n\right)\right)\right)}^{r_{k}}\right],\\ t_{1}\in W_{c_{1}},t_{2},\ldots ,t_{k}\in \overset{M_{1}}{\underset{m=1}{⋃}}W_{c_{m}},i=\bar{1,N}. \end{array}$$

The realization of the statistical estimation of the mixed initial moment function $C_{p_{\xi_{i_{1}}\ldots \xi_{i_{k}}}}\left(t_{1},\ldots ,t_{k}\right)$ of order $p=\sum_{i=1}^{k}r_{i}$ of vector $\Theta_{N}\left(\omega ,t\right)$ is:(28)$$\begin{array}{c} {\hat{C}}_{p_{\xi_{i_{1}}\ldots \xi_{i_{k}}}}\left(t_{1},\ldots ,t_{k}\right)=\frac{1}{M-M_{1}+1}\cdot \sum_{n=0}^{M-M_{1}}\left[\xi_{i_{1_{\omega }}}^{R_{1}}\left(t_{1}+T\left(t_{1},n\right)\right)\cdot \ldots \cdot \xi_{i_{k_{\omega }}}^{R_{k}}\left(t_{k}+T\left(t_{k},n\right)\right)\right],\\ t_{1}\in W_{c_{1}},t_{2},\ldots ,t_{k}\in \overset{M_{1}}{\underset{m=1}{⋃}}W_{c_{m}},i_{i},\ldots i_{k}=\bar{1,N}. \end{array}$$

The realization of the statistical estimation of the mixed central moment function $R_{p_{\xi_{i_{1}}\ldots \xi_{i_{k}}}}\left(t_{1},\ldots ,t_{k}\right)$ of order $p=\sum_{i=1}^{k}r_{i}$ of vector $\Theta_{N}\left(\omega ,t\right)$ is:(29)$$\begin{array}{c} {\hat{R}}_{p_{\xi_{i_{1}}\ldots \xi_{i_{k}}}}\left(t_{1},\ldots ,t_{k}\right)=\frac{1}{M-M_{1}}\cdot \sum_{n=0}^{M-M_{1}}{\left(\xi_{i_{1_{\omega }}}\left(t_{1}+T\left(t_{1},n\right)\right)-{\hat{m}}_{\xi_{i_{1}}}\left(t_{1}+T\left(t_{1},n\right)\right)\right)}^{r_{1}}\cdot \ldots \cdot \\ \cdot \ldots \cdot {\left(\xi_{i_{k_{\omega }}}\left(t_{k}+T\left(t_{k},n\right)\right)-{\hat{m}}_{\xi_{i_{k}}}\left(t_{k}+T\left(t_{k},n\right)\right)\right)}^{r_{k}},\\ t_{1}\in W_{c_{1}},t_{2},\ldots ,t_{k}\in \overset{M_{1}}{\underset{m=1}{⋃}}W_{c_{m}},i_{i},\ldots i_{k}=\bar{1,N}. \end{array}$$

The developed methods of statistical estimation of the compatible probabilistic characteristics of cyclic rhythmically connected random processes allow their joint statistical analysis within the framework of moment functions.

## 5. Experimental Section

In order to verify the developed mathematical model and methods of statistically estimating vector EEG characteristics, a number of measurement experiments were conducted. The OpenBCI platform was used to register the EEG signals. The platform used an 8-channel, 24-bit, low-noise analog-to-digital converter (ADC) ADS1299ADC to record EEG signals (Figure 3). The frequency of data sampling was 250 Hz for each channel. An Ultracortex Mark IV headset (Figure 4) was used to fix the electrodes (Cz, C3, and C4). The electrodes fixed in the headset were dry-typed with Ag-AgCl coating. Each electrode had 12 blunt teeth with a length of 5 mm, which ensured comfort and good contact with the surface of the scalp of the BCI operator.

The OpenBCI GUI utility was used to control the EEG signal recording (Figure 5). The measurement results were recorded on a microSD card directly installed on the board. To process the obtained results, a number of scripts were written in Python using the following auxiliary libraries: sklearn, numpy, scipy, matplotlib, etc.

The experiment essentially consisted of the continuous registration of the vector EEG of the BCI operator, who had to visualize the process of bending and extending the fingers of the palm of the right hand according to the voice commands of the experiment manager. In total, a series of 10 experiments (for 10 different BCI operators) was conducted, each of which consisted of 50 cycles of exposure to the BCI operator.

According to the scheme presented in Figure 1, before applying the methods of evaluating the vector EEG characteristics and the methods of detecting (classifying) the controlling mental influences, it was necessary to conduct preliminary processing of the investigated EEG signals. For such processing, we used Butterworth filters. For the first stage, a notch filter of the 3rd order was used. Its task was to filter the noise of the power grid with a frequency of 50 Hz (60 Hz). The signals before and after the first stage of filtering are shown in Figure 6 and Figure 7.

For the second stage, a bandpass filter of the 5th order was used. In this experiment, the bandwidth of the filter was 1–17 Hz. This allowed us to eliminate all low- and high-frequency noise. The filtered signals, which were ready for the next stages of processing, are shown in Figure 8.

Based on the above-described methods of the statistical evaluation of the probabilistic characteristics of the vector of cyclic rhythmically connected random processes, a statistical evaluation of the mathematical expectation, dispersion, and central and initial moment functions of higher orders was carried out. For all obtained statistical estimates, the Fourier transformation was carried out.

Figure 9 shows a graph of the estimated rhythm function of the vector EEG, which was obtained based on the method of piecewise linear interpolation.

Figure 10, Figure 11, Figure 12, Figure 13, Figure 14, Figure 15, Figure 16, Figure 17, Figure 18, Figure 19, Figure 20 and Figure 21 show graphs of the realizations of the statistical estimates of the probability characteristics of the vector EEG component for the zones of passivity and activity of the BCI operator, which we obtained using methods of statistically estimating the probability characteristics of the vector of cyclic rhythmically connected random processes. In all these figures, the upper, middle, and lower graphs correspond to the first, second, and third EEG channels, respectively.

As shown in the graphs of the statistical estimates of the probability characteristics of the vector EEG in the time and spectral domain, significant differences between the zones of activity and passivity can be visually noted, which indicate the sensitivity of these probability characteristics to the mental control action of the BCI operator. The differences in the structure of the characteristics between the zones of activity and passivity are especially pronounced for the moment functions of the higher-order vector EEG components.

We can also see a significant similarity in the temporal structure of the statistical estimates of the probability characteristics of all components of the vector EEG for the activity zone of the BCI operator, which indicates the commonality of the regularities in these structures for different areas of brain functioning. We also note the possibility of conducting an effective joint statistical assessment of the probability characteristics of the components of the vector EEG and the possibility of coordinated integration of information from different sensors in order to increase the accuracy of detection of the mental control action of the BCI operator.

In general, the above results indicate the adequacy and ability of the developed mathematical model and methods of processing vector EEGs based on the vector of cyclic rhythmically related random processes to reflect the mental controlling action of the BCI operator in its characteristics and the possibility of conducting a high-precision procedure for its detection. However, because there are many such characteristics and there is significant similarity in their temporal and spectral structures, it is necessary to select the most informative characteristics of the vector EEG. This requires the determination of the level of sensitivity of the received characteristic estimates to the mental controlling influence of the BCI operator and selecting those with the highest level of sensitivity. The basis of this procedure is the calculation of the distance between the estimates of the same type of probabilistic characteristics or their spectra in the frequency domain, which correspond to the zones of activity and passivity of the BCI operator. As an example, the results of the calculated mean absolute distances for operators 1 and 2 are given in Table 2 and Table 3.

As can be seen from Table 2 and Table 3, the higher-order moment functions of the components of the vector EEG are the most sensitive to the action of BCI operators, which indicates the expediency of using them as informative characteristics for the detection and classification of the mental control effects of the BCI operator.

In addition to the accuracy of the functioning of BCI systems, an important component of their effectiveness is the amount of computational complexity of the algorithms in BCI systems. Reducing the computational complexity of the algorithms used for the functioning of BCI systems can be achieved by minimizing the dimensions of the vectors of informative features, which are evaluated using the registered vector EEGs. In particular, based on Bessel’s inequality, we propose using the set of the first coefficients of the Fourier-series decomposition (or the set of the first values of the Fourier transform) of the obtained statistical estimates of vector EEGs as a vector of informative features. The number of the first coefficients of the decomposition in the Fourier series (or the number of the first values of the Fourier transform) was chosen in such a way that their contribution to the total energy of the statistical estimate of the moment function was at least 95%. With this approach, the dimensions of the vector of informative features were reduced from 500 (the number of values of the statistical estimate of the moment function of the vector EEG component) to 20 (the number of power spectral density values, which made up at least 95% of the energy of the statistical estimate of the moment function of the vector EEG component).

## 6. Discussion

Compared with the known mathematical models of EEG signals, the new mathematical model enables the study of a much wider class of possible characteristics (multidimensional distribution functions; initial, central and mixed moment functions of higher order for each component of vector EEG; and their respective compatible probabilistic characteristics) of EEG signals, among which the most informative characteristics can be selected. This will increase the accuracy of the detection (classification) of mental control influences of the BCI operators, and will reduce the computational complexity of the algorithms used for the functioning of BCI systems.

As can be seen from Table 1, the new mathematical model of EEG signals registered under the conditions of multiple repetitions of the mental control influences of a BCI operator in the form of a vector of cyclic rhythmically connected random processes has a number of advantages over the previous models. It simultaneously takes into account the stochasticity, cyclicity, variability, and commonality of the rhythm of the vector EEG components. The mathematical model considers the commonality of the rhythmic structures of the components of the vector EEG registered under the conditions of multiple repetitions of the mental control influences of BCI operator. In addition, the clear formal reflection of such commonality makes it possible to increase the accuracy of the detection of control information (control signals) generated by the brain of the operator during their mental control influences, because it is possible to take into account the averaged control information received from different components of vector EEGs. Due to the consideration of the stochastic dependence between the different cycles of the vector EEG and of information about the patterns of changes in the rhythms of the oscillatory process in the vector EEG of the BCI operator, it is possible to obtain more complete, accurate information not only about the mental control influences of the BCI operator, but also about the patterns of training, in particular, the effectiveness of the training of the BCI operator.

The new model has effective statistical means of studying a wide class of characteristics of EEG signals and makes it possible to consider and integrate data obtained from different areas of the BCI operator’s head surface. Therefore, it was taken as the basis of the mathematical modeling of vector EEGs, because this model more fully considers the features of the spatiotemporal structure of the EEG signals registered under the conditions of multiple repetitions of the mental control influences of BCI operators in comparison with existing mathematical models. Additionally, among the advantages of the developed mathematical model of vector EEG is the existence of existing methods of computer simulation of the vector of random signals, which makes it possible to conduct computer simulation experiments for the purpose of testing and optimizing hardware and software for processing vector EEGs in BCI systems [55].

The results of 10 series of experiments with different BCI operators indicated the significant sensitivity of the higher-order moment functions (above the second order) of EEG signals to the mental control action of the BCI operator, which is the basis for their use as informative characteristics when forming a vector of informative features for training and testing classifiers in BCI systems. This result is the basis for supplementing or even replacing the known characteristics of EEG signals, which are mainly characteristics of the first and second order (mathematical expectation, variance, autocorrelation function, and power spectral density function of EEG signals), which are often used in BCI systems, providing one method of increasing the accuracy of the detection and classification of the mental control influences of BCI operators.

The application of Bessel’s inequality in tasks of reducing the dimensionality of the vectors of informative features makes it possible to significantly reduce the computational complexity of the algorithms used in BCI systems, provided that the required level of detection and classification accuracy is maintained. When applying such an approach to reduce the computational complexity of algorithms in BCI systems, provided that the decomposition of informative characteristics of EEG signals is used in other orthogonal non-harmonic bases, we can expect an even stronger reduction in the dimensionality of the vectors of informative features and, accordingly, a larger reduction in computational complexity.

In general, the potential of the developed mathematical model of vector EEGs registered under the conditions of multiple repetitions of the mental control influences of a BCI operator in the form of a vector of cyclic rhythmically connected random processes for the task of building effective BCI systems was not fully obtained in this study. Thus, relevant research in the future should be conducted, in particular in the directions of researching, first, the statistical evaluations of mixed moment functions of vector EEGs; second, the level of stochastic relationships between different cycles of vector EEG; third, methods of computer simulation of vector EEG for the tasks of testing BCI systems and optimizing algorithms of their functioning; and, finally, the development of optimal algorithms for the detection and classification of the controlling mental influences of the BCI operator. As informative characteristics, the probabilistic characteristics of the vector of cyclic rhythmically connected random processes can be used, as shown in this paper.

The results obtained in the study are the first important stage in the future comprehensive study of the characteristics of EEG signals. The next stage is conducting significantly larger experiments both in terms of the number of BCI operators and the volume of the set of investigated characteristics of EEG signals. This comprehensive study is planned to be implemented within the framework of several separate projects.

## 7. Conclusions

A new mathematical model of vector EEGs registered under the conditions of multiple repetitions of the mental control influences of a BCI operator in the form of the vector of cyclic rhythmically connected random processes was substantiated, which, due to considering the stochasticity and cyclicity, variability, and the commonality of the rhythm of multidimensional distribution functions as well as the initial, central, and mixed moment functions of the investigated signals, has a number of advantages over the prior models. As the mathematical model of the vector EEG in the form of vector of cyclic rhythmically connected random processes integrally describes not only the signal from one local area of the surface of the human head, but also their synchronously registered aggregate, taking into account their common rhythmic structure, on the basis of this model, it is possible to carry out a compatible statistical analysis of signals. This makes it possible to consider and integrate data obtained from different areas of the surface of the BCI operator’s head in order to increase the accuracy of the detection of the mental control action of the BCI operator.

The methods of statistical processing of vector EEGs registered under the conditions of multiple repetitions of the mental control influences of BCI operators were substantiated, which consisted of the statistical evaluation of its probabilistic characteristics. In contrast to the known methods of statistical processing of EEGs, which are based on the statistical averaging of an ensemble of EEG realizations, the developed methods enable the estimation of the stochastic dependence between the different cycles of vector EEGs, which correspond to different experiments in a single series, which contributes to increasing the accuracy of the functioning of BCI systems. The obtained results indicated the adequacy and ability of the developed mathematical model and methods for vector EEG processing based on the vector of cyclic rhythmically related random processes to reflect the mental controlling actions of the BCI operator in its characteristics and the possibility of conducting a high-precision procedure for their detection.

The use of higher-order moment functions (above the second order) and their spectral images in the frequency domain as informative characteristics in BCI systems was substantiated. Their significant sensitivity to the mental controlling influence of the BCI operator was experimentally established.

In order to reduce the computational complexity of the functioning algorithms of BCI systems, the dimensions of the vectors of informative features was minimized. Namely, it was presented in the form of a set of the first coefficients of the Fourier-series decomposition (or the set of the first values of the Fourier transform) of the obtained statistical estimates of the vector EEG components. It was experimentally established that only the first 20 values of the Fourier transform of the moment function estimation of the vector EEG component were needed to form the vector of informative features in the BCI system, because, in total, these spectral components made up at least 95% of the total energy of the corresponding statistical estimate of the moment function of the vector EEG components.

## Figures and Tables

**Figure 1 sensors-23-00760-f001:**
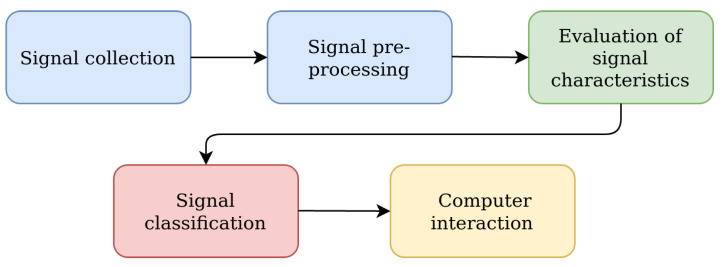
Main stages of signal processing by a BCI system.

**Figure 2 sensors-23-00760-f002:**
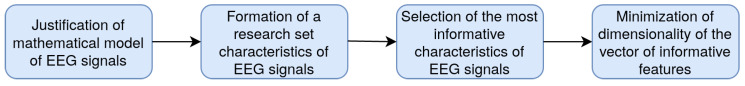
Illustration of a conceptual approach to the study of the characteristics of EEG signals.

**Figure 3 sensors-23-00760-f003:**
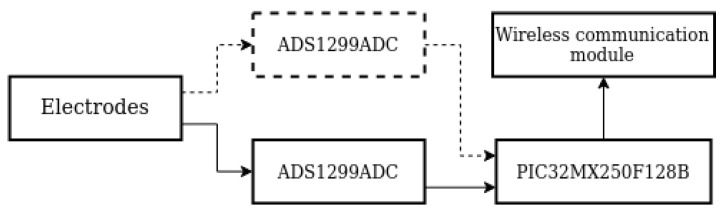
Block diagram of the OpenBCI platform.

**Figure 4 sensors-23-00760-f004:**
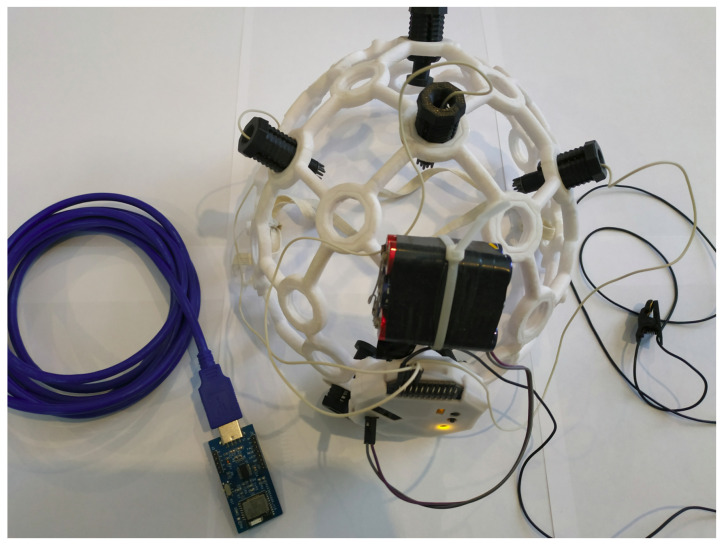
The open source brain–computer Interface platform.

**Figure 5 sensors-23-00760-f005:**
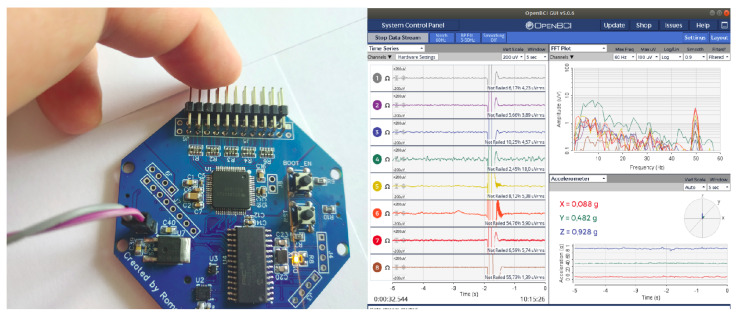
The OpenBCI GUI.

**Figure 6 sensors-23-00760-f006:**
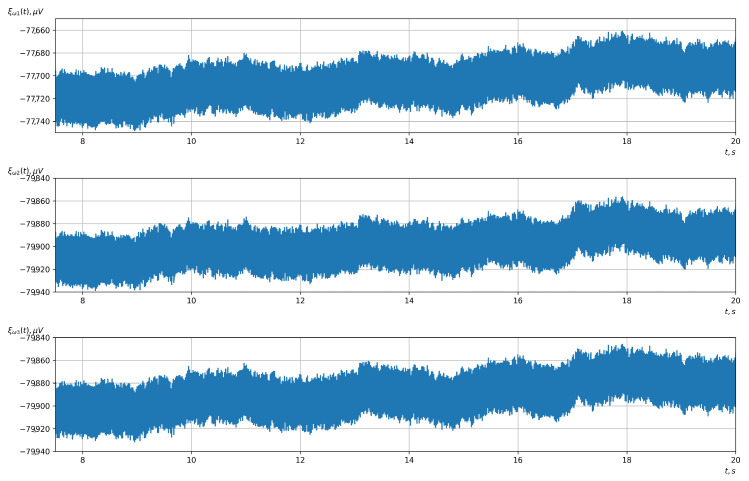
The signals after the analog-to-digital converter, recorded by the OpenBCI platform (the upper, middle, and lower graphs correspond to the 1st, 2nd, and 3rd EEG channels, respectively).

**Figure 7 sensors-23-00760-f007:**
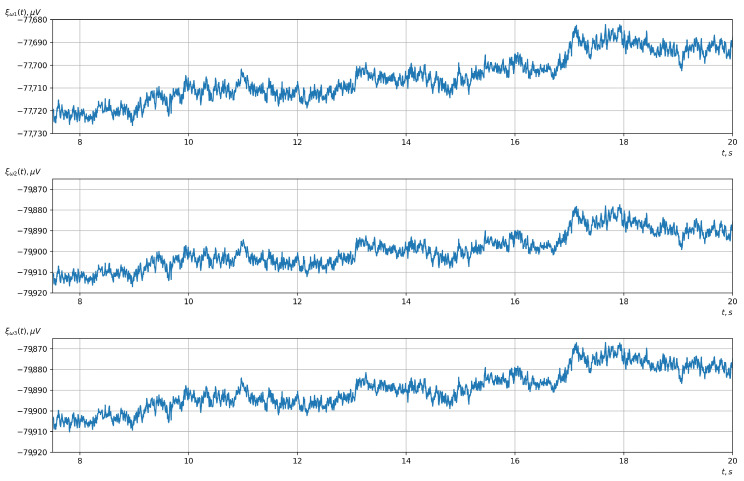
The signals after the first stage of filtering with a 50 Hz notch filter (the upper, middle, and lower graphs correspond to the 1st, 2nd, and 3rd EEG channels, respectively).

**Figure 8 sensors-23-00760-f008:**
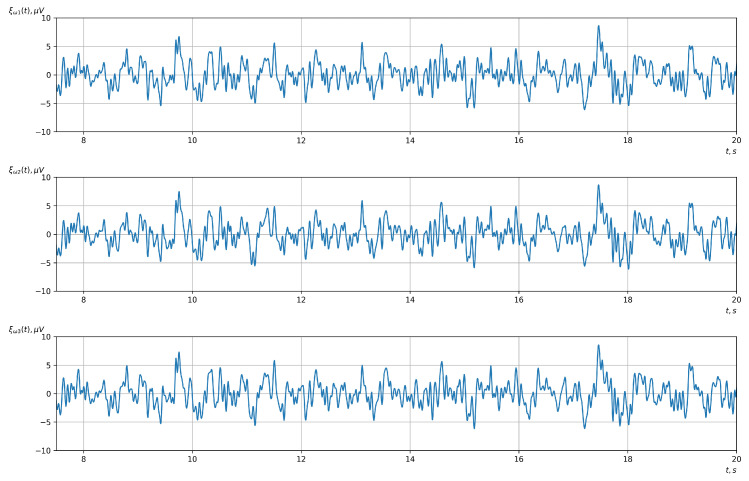
The signals after the second stage of filtering with a bandpass filter (the upper, middle, and lower graphs correspond to the 1st, 2nd, and 3rd EEG channels, respectively).

**Figure 9 sensors-23-00760-f009:**
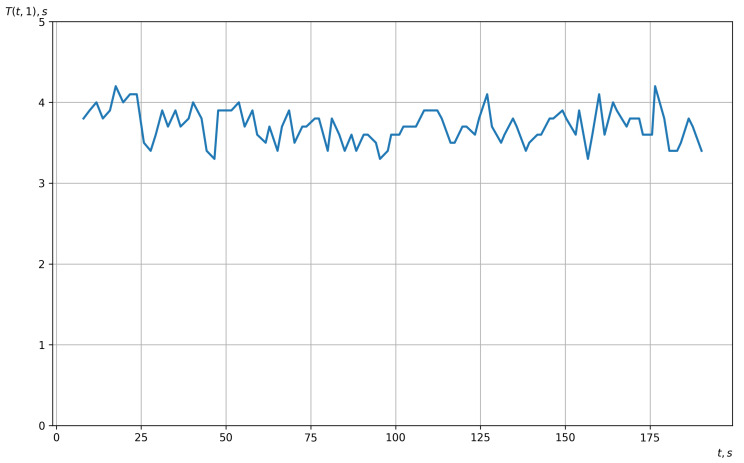
Graph of the estimation of the vector EEG rhythm function.

**Figure 10 sensors-23-00760-f010:**
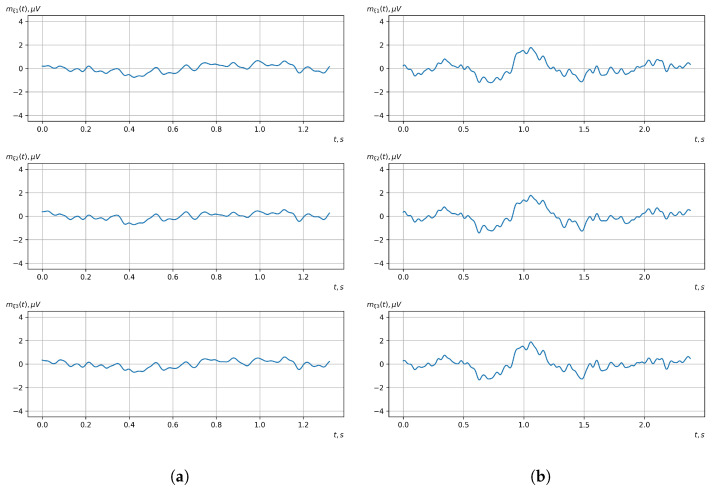
Graphs of realizations of statistical estimates of mathematical expectations of vector EEG components: (**a**) zone of passivity (lack of action) of the BCI operator; (**b**) zone of activity of the BCI operator.

**Figure 11 sensors-23-00760-f011:**
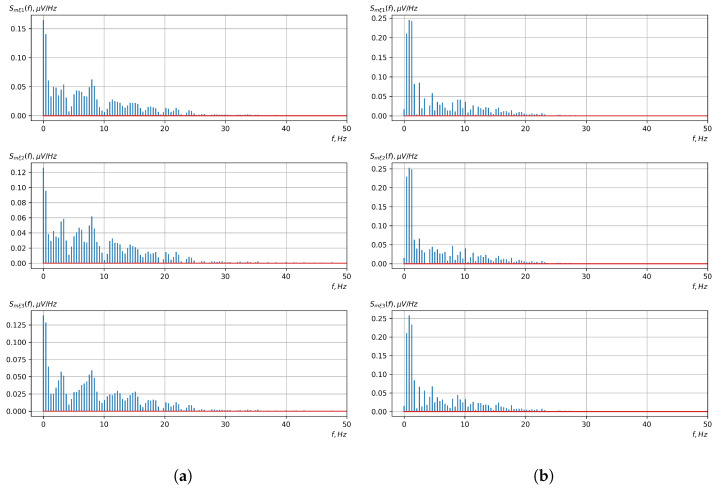
Graphs of Fourier transforms of realizations of of statistical estimates of mathematical expectations of vector EEG components: (**a**) zone of passivity (lack of action) of the BCI operator; (**b**) zone of activity of the BCI operator.

**Figure 12 sensors-23-00760-f012:**
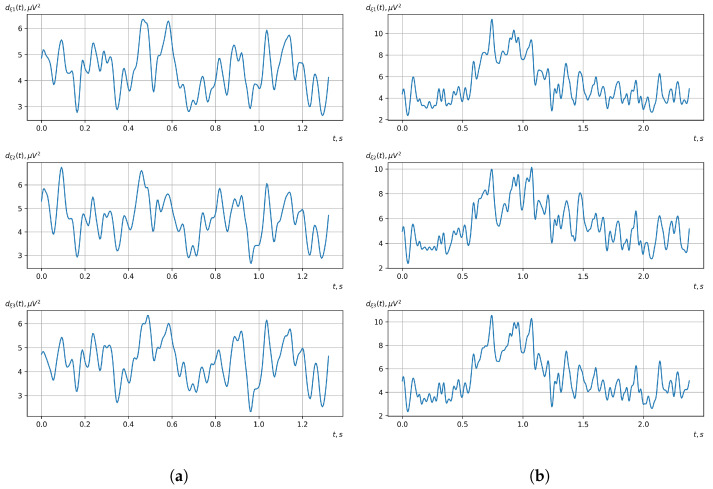
Graphs of realizations of statistical estimates of the initial moment functions of the second-order vector EEG components: (**a**) zone of passivity (lack of action) of the BCI operator; (**b**) zone of activity of the BCI operator.

**Figure 13 sensors-23-00760-f013:**
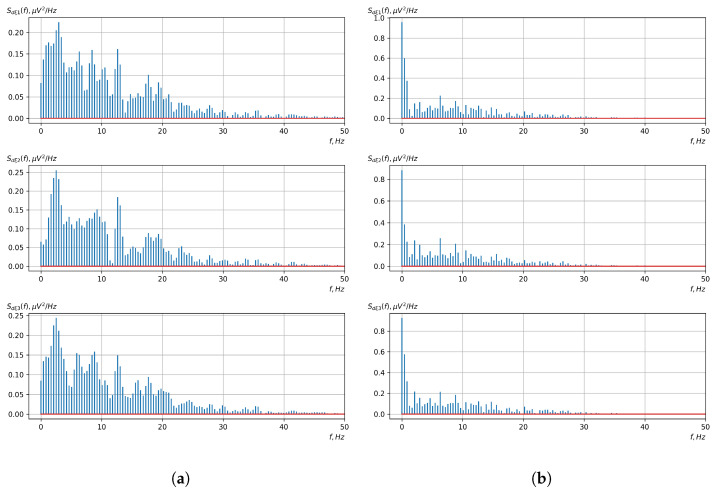
Graphs of Fourier transforms of realizations of statistical estimates of the initial moment functions of the second-order vector EEG components: (**a**) zone of passivity (lack of action) of the BCI operator; (**b**) zone of activity of the BCI operator.

**Figure 14 sensors-23-00760-f014:**
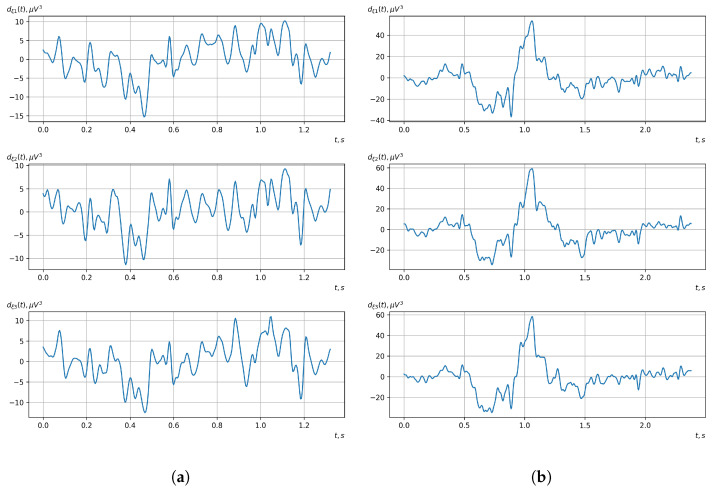
Graphs of realizations of statistical estimates of the initial moment functions of the third-order vector EEG components: (**a**) zone of passivity (lack of action) of the BCI operator; (**b**) zone of activity of the BCI operator.

**Figure 15 sensors-23-00760-f015:**
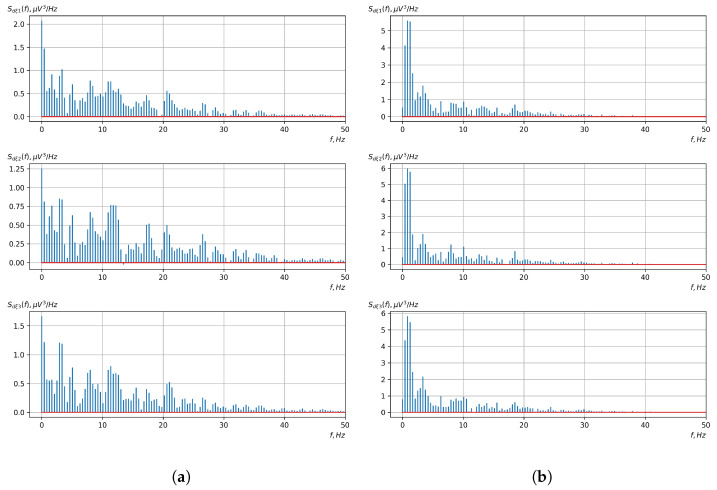
Graphs of Fourier transforms of realizations of statistical estimates of the initial moment functions of the third-order vector EEG components: (**a**) zone of passivity (lack of action) of the BCI operator; (**b**) zone of activity of the BCI operator.

**Figure 16 sensors-23-00760-f016:**
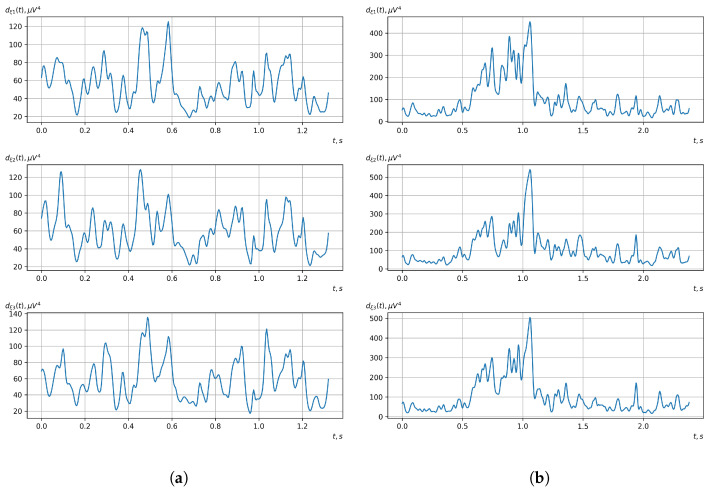
Graphs of realizations of statistical estimates of the initial moment functions of the fourth-order vector EEG components: (**a**) zone of passivity (lack of action) of the BCI operator; (**b**) zone of activity of the BCI operator.

**Figure 17 sensors-23-00760-f017:**
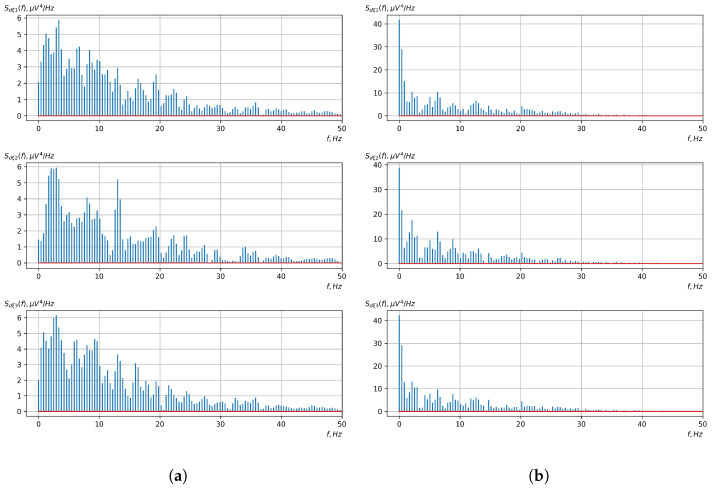
Graphs of Fourier transforms of realizations of statistical estimates of the initial moment functions of the fourth-order vector EEG component: (**a**) zone of passivity (lack of action) of the BCI operator; (**b**) zone of activity of the BCI operator.

**Figure 18 sensors-23-00760-f018:**
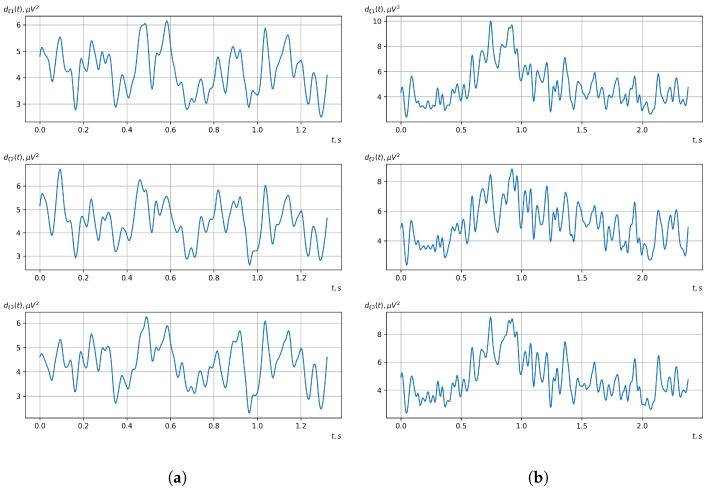
Graphs of realizations of statistical estimates of the dispersions of vector EEG components: (**a**) zone of passivity (lack of action) of the BCI operator; (**b**) zone of activity of the BCI operator.

**Figure 19 sensors-23-00760-f019:**
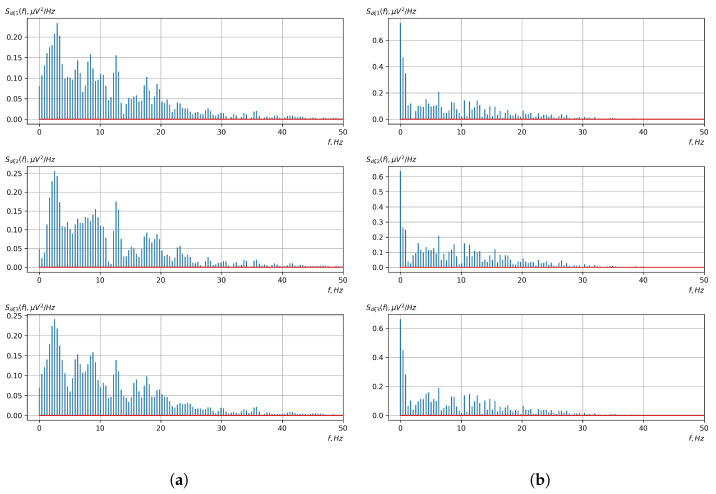
Graphs of Fourier transforms of realizations of statistical estimates of the dispersions of vector EEG components: (**a**) zone of passivity (lack of action) of the BCI operator; (**b**) zone of activity of the BCI operator.

**Figure 20 sensors-23-00760-f020:**
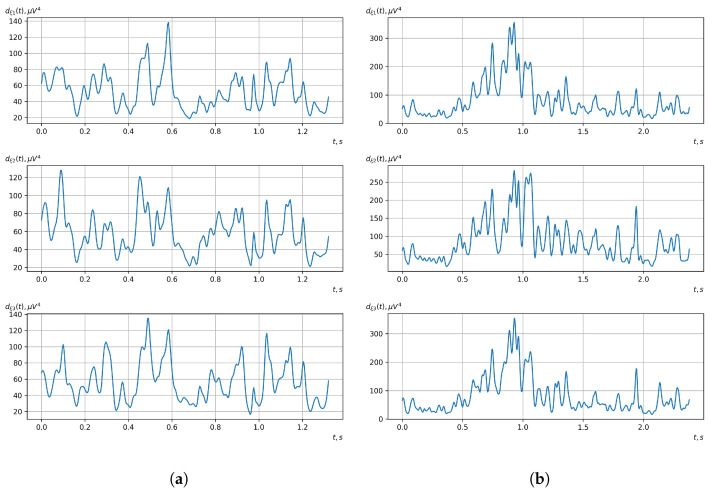
Graphs of realizations of statistical estimates of the central moment functions of the fourth order vector EEG components: (**a**) zone of passivity (lack of action) of the BCI operator; (**b**) zone of activity of the BCI operator.

**Figure 21 sensors-23-00760-f021:**
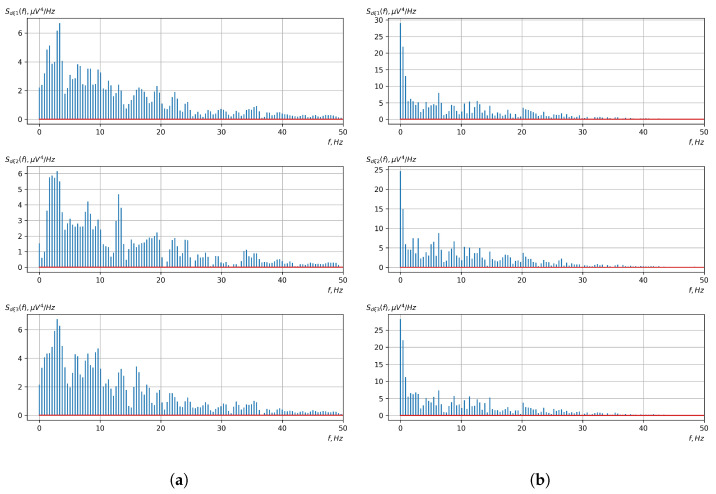
Graphs of Fourier transforms of realizations of statistical estimates of the central moment functions of the fourth-order vector EEG components: (**a**) zone of passivity (lack of action) of the BCI operator; (**b**) zone of activity of the BCI operator.

**Table 1 sensors-23-00760-t001:** Comparative characteristics of existing and proposed mathematical models for EEG signals.

	Previous EEG Mathematical Models							New Model
	Deterministic models [22,23,24,25,26,27,28,29,30,31]	Stationary random process in wide sense [7,32]	Stationary sequence of autoregression [34,35]	Stationary linear random process [36,37,38]	Piecewise stationary random process [36,43,44]	Vector of stationary and stationary connected random processes [42]	Cyclostationary correlated random process [45]	The vector of cyclic rhythmically connected random processes
Considers random nature of EEG	-	+	+	+	+	+	+	+
Considers cyclical structure of probabilistic characteristics of EEG	-	-	-	-	-	-	+	+
Considers compatible probabilistic characteristics of various components in vector EEG	-	-	-	-	-	+	-	+
Considers irregularity (variability)in vector EEG rhythm	-	-	-	-	+	-	-	+
Considers commonality of rhythmic structure for all components of vector EEG	-	-	-	-	-	-	-	+
Considers presence of zonal structure of vector EEG cycle	+	-	-	-	+	-	-	+
Considers moment functions of higher order of vector EEG	-	-	-	+	-	+	-	+
Considers stochastic dependence between different cycles of vector EEG	-	-	-	-	-	-	-	+

“+”–takes into account (displays). “-”–does not take into account (does not display).

**Table 2 sensors-23-00760-t002:** Results of calculated mean absolute distances between estimates of same type of characteristics of vector EEG components for zones of activity and passivity for operator 1.

	Mean Absolute Distances between Moment Functions			Mean Absolute Distances between Spectra of Moment Functions		
Electrode	$C_{3}$	$C_{Z}$	$C_{4}$	$C_{3}$	$C_{Z}$	$C_{4}$
Initial moments of fourth order	90.5877	87.0864	90.8943	3.3048	3.8014	3.3564
Central moment functions of fourth order	66.2054	59.4049	64.1556	2.4268	2.2709	2.4801
Initial moments of third order	14.3101	13.9428	14.2710	0.5633	0.6079	0.6002
Initial moments of second order	2.2175	2.0566	2.0953	0.0686	0.0642	0.0659
Variance	1.7699	1.5484	1.5843	0.0643	0.0660	0.0668
Mathematical expectation	0.6431	0.6415	0.6327	0.0256	0.0247	0.0250

**Table 3 sensors-23-00760-t003:** Results of calculated mean absolute distances between estimates of same type of characteristics of vector EEG components for zones of activity and passivity for operator 2.

	Mean Absolute Distances between Moment Functions			Mean Absolute Distances between Spectra of Moment Functions		
Electrode	$C_{3}$	$C_{Z}$	$C_{4}$	$C_{3}$	$C_{Z}$	$C_{4}$
Initial moments of fourth order	28.5779	30.6315	32.1608	1.3689	1.1641	1.3366
Central moment functions of fourth order	26.4678	28.2864	28.0672	2.4268	2.2709	2.4801
Initial moments of third order	6.8095	7.4268	8.0563	0.3508	0.2961	0.3211
Initial moments of second order	1.0194	1.0798	1.1567	0.0439	0.0444	0.0460
Variance	1.0209	1.0457	1.1054	0.0413	0.0440	0.0449
Mathematical expectation	0.3804	0.3934	0.4165	0.0192	0.0181	0.0176

## Data Availability

Not applicable.

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
