# Peer review of "Advanced Modeling and Signal Processing Methods in Brain–Computer Interfaces Based on a Vector of Cyclic Rhythmically Connected Random Processes"

_sensors, 2023, doi:10.3390/s23020760_

Round 1

Reviewer 1 Report

The proposed study describes the mathematical model of vector EEG in the form of a vector of cyclic rhythmically connected random processes for using in Brain-Computer Interfaces. The paper is interesting but is not well-written and has serious drawbacks:

1 - Please use reference for the mentioned methods in the text of page 2 (lines 72 to 79) and page 3 (lines 80 to 90).

2 - The structuring and organization of this article is not good.

3 - For better explanation, Section 3 (results) should be divided into two sections for example mathematical model of EEG signals and results of statistical of EEG signals.

4 - How many data was used to classify in BCI?

5 - Please specify in figures 5, 6 and 7 which of the subplots corresponds to which one of the EEG channels.

6 - The results of the paper are not clearly presented. The authors should compare their results with related papers.

7 - More discussion on results is needed.

Reviewer 2 Report

The present study developed a mathematical model for signal processing methods in brain-computer interfaces. The manuscript needs a major revision based on novelty, findings, contributions, etc. My comment and suggestions are as follows:

1.      Try to avoid abbreviations in the abstract and rewrite the abstract with proper novelty and findings with some numerical conclusions.

2.      Introduction section is not written properly. In this section, provide the objective, motivations, research gaps, contributions, and complexity of your study with some subsections.

3.      Try to avoid bulky references in the related work section. Each research had unique findings, try to illustrate that or remove them if it is not essential.

4.      Provide an author(s) contribution table in this section to show the novelty of this study compared to existing literature.

5.      What will you assume to make this model? Describe each assumption with proper references.

6.      Several mathematical terms and symbols are used throughout the manuscript, along with several abbreviations. Thus, please provide a section for Notations and abbreviations separately.  

7.      Conclusion section should be rewritten consequently instead of numbering and provide limitations of this study along with some future extensions. Future extensions should be properly referenced.

8.      Please describe the managerial implications of this study before the conclusion in a separate section.

9.      Remove all “I,” “We,” and “Our” throughout the manuscript and rewrite the sentence. Moreover, an English language check is required for whole manuscript.

Round 2

Author Response

Dear Reviewer,
the updated version of the article has been uploaded to the system.

Reviewer 2 Report

Revision is still required as the authors neglected some of my suggestions. It was suggested to remove bulky references and illustrate the findings of each reference properly but the authors used bulky references in several places, which indicates that the authors are careless about revision, moreover, it was advised to put a notation section in which all notations should be discussed properly, the author adds Abbreviations and notation section but all notations are not discussed here, for example, R, Z, omega, gamma and many more. It is impossible to understand which part was edited by the authors, as per my knowledge during the submission of the revised version the authors should use track change methods or highlights the modified part in the manuscript such that it can be understandable.
